# A New Computer Model for Evaluating the Selective Binding Affinity of Phenylalkylamines to T-Type Ca^2+^ Channels

**DOI:** 10.3390/ph14020141

**Published:** 2021-02-10

**Authors:** You Lu, Ming Li

**Affiliations:** 1Center for Aging, School of Medicine, Tulane University, New Orleans, LA 70112, USA; ylu6@tulane.edu; 2Department of Physiology, School of Medicine, Tulane University, New Orleans, LA 70112, USA

**Keywords:** T-type calcium channel blocker, homology modeling, computer-aid drug design, virtual drug screening, L-type calcium channel

## Abstract

To establish a computer model for evaluating the binding affinity of phenylalkylamines (PAAs) to T-type Ca^2+^ channels (TCCs), we created new homology models for both TCCs and a L-type calcium channel (LCC). We found that PAAs have a high affinity for domains I and IV of TCCs and a low affinity for domains III and IV of the LCC. Therefore, they should be considered as favorable candidates for TCC blockers. The new homology models were validated with some commonly recognized TCC blockers that are well characterized. Additionally, examples of the TCC blockers created were also evaluated using these models.

## 1. Introduction

As the only type of voltage-gated Ca^2+^ channels that are activated at or near resting membrane potentials, T-type Ca^2+^ channels (TCCs) play an important role in regulating [Ca^2+^]_i_ homeostasis in a variety of tissues, including pancreatic β-cells and tumor cells [1,2,3,4]. Therefore, TCC antagonists could be potentially useful for the treatment of chronic diseases associated with Ca^2+^ dysregulation [5,6,7]. For this reason, it is imperative to develop more selective TCC antagonists for prospective clinical applications. Since many existing TCC blockers, such as mibefradil, also show inhibitory effects on L-type calcium channels (LCCs), the most important task in developing new TCC blockers is to enhance their selectivity to TCCs over LCCs. To achieve this, we established TCC–phenylalkylamine interaction models based on the specific amino acid sequences in the P-loop of TCCs, and α_1_C LCCs for characterizing the drug molecules’ affinities for TCCs and LCCs, respectively.

TCCs have a close evolutional relationship with LCCs. A recent report from a cryo-electron microscopy study reveals that the frame of the α_1_G (Ca_v_3.1) pore domain structure is similar to that of α_1_S (Ca_v_1.1) [8]. This similarity allowed us to confidently adopt the global structure of the calcium channel Ca_v_Ab model, constructed based upon Arcobacter butzleri crystallization [9], in the establishment of our TCC model. One of the most remarkable differences between all types of TCCs and LCCs is a lysine residue located adjacent to the critical glutamic acid/aspartic acid residue in domain III. The existence of a positively charged lysine (K^3p49^) may swing the aspartic acid (D^3p50^) away from the center of the calcium filter and change the preferred calcium ion and drug binding sites from domains III and IV for LCCs to domains I and IV for TCCs. Therefore, we used the ZMM molecule modeling program [10,11,12] to create four-domain TCC models, in which the binding affinities of drugs to TCCs and α_1_C LCC were determined by scoring their free energy in binding to the channels [13].

The new TCC models are adopted from a drug–protein interaction framework for modeling Ca_v_Ab blocking by phenylalkylamines (PAAs) [9]. This is rational because many TCC blockers are PAAs or their derivatives, and because PAAs block Ca_v_Ab [9,14]. It is proposed that PAAs bind to LCCs in an inverse V-shaped configuration, with the ammonium group towards the P-helices, and the nitrile group bound to the calcium ion coordinated by the selectivity filter glutamates in domains III and IV of the LCC [15]. We reason that this is also true for TCC blockers, except that the calcium ion is coordinated in the cavity between domains I and IV, since the depolarization confirmed that the pore domains of Ca_V_3.1 and Ca_V_1.1 are superimposed [8]. As a result, the two rings of the flexible PAA molecule [15] will make hydrogen bonding contacts with the mobile side chains of relevant amino acids from domains I and IV of TCCs. This strategy allowed us to create computer models for simulating the interactions between drugs and channel receptors for LCC and TCCs, respectively.

## 2. Results

### 2.1. Homology Modeling of TCCs and α_1_C LCC

Using the bacterial calcium Ca_v_Ab open channel 3D structure as the input, ZMM generated the first template of the calcium channel, which was then modified with S5-P-loop-S6 segments of α_1_C, and α_1_G, α_1_H, and α_1_I (Table 1) to create corresponding protein structures of Ca_v_1.2 LCC and Ca_v_3.1, Ca_v_3.2, and Ca_v_3.3 TCCs, respectively (Figure 1A–D). For cross-validation of ZMM-generated 3D structure models, we also performed ab initio modeling of α_1_C and α_1_G calcium channels. Since there is a considerable overlap of PAA inhibition between LCC and TCCs [9], the allosteric structures of LCC and TCCs are more likely to be similar. Comparing two different homology modeling tools, ZMM generates more consistent 3D models of the domain III S5-P-loop-S6 segment of α_1_C and α_1_G (Figure 1E,F) than the ab initio method (Figure 1G,H).

### 2.2. Further P-Loop Remodeling of TCCs

After determining the globe structure of TCC 3D models, we focused on the variability of the P-loop structure, which is the major drug–ligand interaction segment. The Rosetta P-loop remodeling module [17] was utilized to estimate the variability of P-loop 3D structures on every domain of TCCs. After inputting a perturbation to the original structure, the remodeling process was conducted by sampling the possible locations of a given length of an amino acid sequence in three-dimensional space. Using the ZMM generated α_1_G structure as the reference, the energy-based clustering method [18] was used to determine the P-loop remodeling results with the lowest root-mean-square-displacement (RMSD) score. We found that for α_1_G, domain II had a clear variation between two different homology modeling methods in sample sizes 500 and 20,000 (Figure 2). It showed that the central P-loop helix segment is in the horizontal position rather than the diagonal position found in other domains. As a result, the selectivity-determining glutamic acids E^2p50^ may have a larger vertical distance from other glutamic acids/aspartic acids (E^1p50^, D^3p50^, and D^4p50^) in α_1_G TCCs. This may exclude glutamic acid E^2p50^ as a Ca^2+^ binding candidate, leaving the Ca^2+^ to bind either E^1p50^ to D^4p50^ or D^3p50^ to D^4p50^ in TCCs. Additionally, to validate the normality of the remodeling data, we conducted a nonparametric test for the α_1_G P-loop remodeling data and confirmed that all the sampling processes (500 and 20,000) came from the same distribution (see Appendix A for statistical results).

### 2.3. Local Electrostatic Potentials of the Selective P-Loop of TCC Domains and the Impact of K^3p49^

A previous study indicated that when a calcium ion enters the selectivity filter region of a LCC, it binds to the selectivity-determining glutamic acids (E^3p50^, E^4p50^) in domains III and IV [15]. Consequently, the phenylalkylamine molecules will bind to domains III and IV due to the interaction between the nitrile nitrogen and Ca^2+^ [15]. In contrast, all TCCs have a lysine (K^3p49^) located at the 5′ end adjacent to D^3p50^ in domain III. It is reported that the replacement of lysine (K^3p49^) with Phe or Gly causes the activation curve to shift to the right [8], which indicates that the lysine in the position adjacent to aspartic acid (D^3p50^) plays a significant role in the kinetic/dynamic mechanism of Ca^2+^ interaction with the inner environment of the central cavity of TCCs. This positively charged lysine alters the negative charge field distribution of D^3p50^ to attract Ca^2+^ (Appendix A). It is possible that the lysine (K^3p49^) swings aspartic acid (D^3p50^) away from the original Ca^2+^ binding position, thus causing the Ca^2+^ to bind glutamic acid or aspartic acid in other domains, probably to domains I and IV since domain II has a configuration deviation. As a result, the phenylalkylamine may also switch its binding region from domains III and IV to domains I and IV.

To determine the effect of lysine (K^3p49^) on overall electrostatic potential (E) for given TCC homology models, we calculated the electrostatic potential (E) for the tailed P-loop of domain I to IV. Table 1 shows that the combined electrostatic potential (E_coul_) becomes more negative in each domain as the number of testing amino acids is reduced from seven to five. In TCCs, E_coul_ for domain I is more negative than that for domain IV in the five amino acid-reduced sequence, indicating a possible switching of the Ca^2+^ binding site from domains III and IV to domains I and IV.

We used Coulomb’s electric force equation to quantitively analyze the influence of lysine on the electrical attraction force between Ca^2+^ and aspartic acid (D^3p50^). According to the equation in Section 4.2, lysine (K^3p49^) has the least effect on the Ca^2+^-D^3p50^ attraction when K^3p49^ is located on the opposite side of the Ca^2+^ and when D^3p50^ is at the center. When a = 4.3 Å and b = 3.8 Å [8] (calculated in PyMOL.2.3.3 for ZMM results), Coulomb’s force equation (found in Section 4) yields: F_(Ca, D, attraction)_ = 2.489 × 10^−9^ N and F_(Ca, K, repellent)_ = 0.702 × 10^−9^ N; thus, lysine, at a minimum, reduces the attraction force between Ca^2+^ and aspartic acid by more than 28%. The attraction force between Ca^2+^ and D^3p50^ will reduce further or reverse into a repellent force as the distance from the Ca^2+^ to K^3p49^ decreases; therefore, the preferred binding position of Ca^2+^ will likely be switched to domains I and IV in TCCs. This limits the PAA binding region on TCCs to domains I and IV. We could use the amino acid structure of domains I and IV to evaluate the affinities of the PAAs (and their derivatives) for TCCs (using models established for α_1_G, α_1_H, and α_1_I) and use the amino acid structure of domains III and IV for evaluating their binding affinities to LCC (using the model of α_1_C).

### 2.4. Model Predictions and Vina Screening Output of Some Current T-Type Ca^2+^ Channel Blockers

Mibefradil is reported to have an inhibitory effect on both LCC and TCCs [20]. The α_1_C model predicts that one hydrogen atom from the nitrogen (N_3_) on the cyclopentadiene connects to methionine (M^4i27^) on domain IV of α_1_C LCC, as shown in Figure 3A,B. This is consistent with the prediction of another model in a previous study [9]. In contrast, the α_1_C homology model does not predict that hydrogen bonds to NNC 55-0396. NNC 55-0395 inhibits both L- and T-type calcium channels [14]. The α_1_C homology model predicts that NNC 55-0395 has one hydrogen bond that connects nitrogen (N_3_) to the glycine (G^3p49^) on P-loop domain IV. For NNC 55-0397, the α_1_C homology model also predicts that RO 40-5966, a hydrolyzed metabolite of mibefradil [20], has one hydrogen atom from the hydroxy group of the benzene ring bound to the glycine (G^4p49^) at domain IV. No hydrogen bond has been found between the LCC and the TCC blocker SKF-96365.

Using TCCs as templates, we have revealed some current TCC blockers of α_1_G, α_1_H, and α_1_I. The α_1_G model predicts that the fluorine atom from the compound NNC 55-0395 forms a halogen bond to glycine (G^1p51^) in domain I. Our α_1_G model also predicts a binding site of NNC 55-0396 to asparagine (N^li20^) in domain I (Figure 3C,D). For NNC 55-0397, the fluorine atom from the compound forms a halogen bond with valine (V^1p46^) at the P-loop of domain I. For mibefradil, one oxygen atom from the side chain of the compound forms hydrogen bonds with asparagine (N^1o4^) at S5 of domain I. For RO 40-5966, one hydrogen atom from the nitrogen (N_3_) on the cyclopentadiene forms a hydrogen bond with alanine (A^1i27^) in domain I. For SKF-96365, the center oxygen atom forms a hydrogen bond with asparagine (N^1o4^) at S5 of domain I. 

Our α_1_H model predicts that one hydrogen atom from the nitrogen (N_3_) on the cyclopentadiene of NNC 55-0395 interacts with valine (V^1p46^) at the P-loop of domain I to form a bond. The fluorine atom from NNC 55-0396 forms a halogen bond with isoleucine (I^1i8^) from the α_1_H S6 of domain I. For NNC 55-0397, one hydrogen atom from the nitrogen (N_3_) on the cyclopentadiene forms a bond to asparagine (N^4p51^) at the P-loop of domain IV. The hydrogen atom from the nitrogen (N_3_) on the cyclopentadiene of mibefradil finds asparagine (N^1o4^) to form a bond at S5 of domain I. For RO 40-5966, the fluorine atom from the compound forms a halogen bond with histidine (H^4i29^) at S6 of domain IV. One oxygen atom from the side chain of SKF-96365 forms a hydrogen bond with asparagine (N^1o4^) at S5 of domain I.

Our α_1_I homology model predicts that NNC 55-0395 forms a halogen bond between the fluorine atom from the compound and isoleucine (I^1i8^) at S6 of domain I. The hydrogen atom from NNC 55-0396 forms a bond to asparagine (N^1o4^) at S5 of domain I. For NNC 55-0397, one hydrogen atom from the nitrogen (N_3_) on the cyclopentadiene interacts with valine (V^1p46^) at the P-loop of domain I to form a bond. For mibefradil, there is a halogen bond formed between a fluorine atom from the compound and a hydrogen atom from asparagine (N^4p53^) at the P-loop of domain IV. For RO 40-5966, one hydrogen atom from the ammonia on the cyclopentadiene interacts with asparagine (N^1o4^) at S5 of domain I to form a bond. Our model does not predict the hydrogen bond formed when docking SKF-96365 to α_1_I. 

A comparison of the predicted binding affinity K_d_ and experimental measurements of IC_50_ for given TCC blockers are listed in Appendix A. Table 2 summarizes the predicted binding affinity results for all existing TCC blockers.

### 2.5. Evaluation of New Compounds

The Vina [13] models were employed for evaluating the binding affinity of the testing compounds. We randomly selected 300,000 compounds from PubChem and used these as the database to train our recurrent neural networks (RNNs) [21] with the given compound properties. 

After performing virtual screening, we found that the compounds TC 7, TC 4, and TC 2 satisfied our screening criteria for α_1_G, α_1_H, and α_1_I, respectively. Compound TC 7 has the highest binding affinity, as well as a lower (water–octanol partition coefficient) logP and a higher Quantitative Estimation of Drug-likeness (QED) than existing TCC blockers. The 3D binding plots between TC 7 and α_1_G are shown in Appendix A. The predicted binding affinities between existing TCC blockers and screened compounds on TCCs and α_1_C LCC are shown in Figure 4. Our results show that these screened compounds have smaller logP and Synthetic Accessibility Scores (SAS) but larger QED values than those of selected TCC blockers (as seen in Table 3). More structures and chemical properties for the 13 identified compounds can be found in Appendix A.

## 3. Discussion

A TCC (Ca_v_3.1) 3D structure has already been modeled with cryo-electron microscopy [8]; however, this structure is constructed based on a splice variant containing a deletion of 133 amino acids within the I-II linker. Electrophysiological characterization of these variants (Ca_v_3.1-Δ8b) shows 1.5-2-fold conductance increases when compared with the full-length form in human and rat preparations. Both activation and steady-state inactivation curves are shifted in the human preparation [8]. In addition, the pore diameter estimated from Ca_v_3.1-Δ8b is smaller than the biophysical measurement [8]. These alterations in TCC electrophysiological properties suggest that the conformation of the cryo-electron microscopy structure is not the same as the full-length Ca_v_3.1 TCC. Therefore, the 3D structure of Ca_v_3.1-Δ8b may not be the most suitable template for the general modeling of TCCs, especially for PAA binding, which is highly dependent on the position of Ca^2+^ interacting with the selectivity filter of TCCs. In this study, we chose to use Ca_v_Ab as the model template since this channel is blocked by PAA and therefore is suitable for establishing a model for evaluating PAAs that inhibit TCCs selectively over LCCs.

Increasing evidence indicates the pathological role of TCCs in the progression of different diseases [6]. It is crucial to develop selective TCC blockers to establish new treatments for these diseases. Unfortunately, lacking the TCC X-ray crystallization structure hampers the progress of creating new TCC blockers. In practice, it is very difficult to find or design a compound that selectively blocks TCCs but not LCCs, since most current TCC blockers exhibit a certain level of inhibitory effects on LCCs. For example, mibefradil, the first launched TCC inhibitor, was quickly recognized to cross inhibit LCC [20]. Here, we provide a new strategy by which the specificity of candidate compounds for binding TCCs but not LCCs can be pre-screened with new computer-based models. This is desirable for designing and developing compounds that more selectively block TCCs than LCCs. 

We chose to build models for the interaction between TCCs and PAAs since the binding mechanism of these compounds to LCCs has been studied extensively [9,15,22,23,24,25,26]. Based upon the critical single amino acid lysine (K^3p49^) difference between LCCs and TCCs, we have created a strategy that can distinguish the affinity of PAAs to TCCs and LCCs, respectively. The models have been validated by measuring the affinities of existing TCC blockers to LCCs and TCCs. We also used these models for evaluating the specificities of novel PAAs and phenylalkylamine derivatives in terms of their binding affinities to TCCs and LCCs.

Using ZMM, we simultaneously created α_1_G, α_1_H, and α_1_I TCC and a_1_C LCC structures with four domains, each containing three segments: segment 5, a P-loop, and segment 6. In contrast, the ab initio modeling method failed to produce a suitable calcium channel structure compared to ZMM.

The selectivity of the calcium channel is dependent on the critical glutamate residues located in the selectivity filter of the P-loop of the α_1_ subunit in each domain. In this region, there are two negatively charged glutamic acid residues likely to attract one Ca^2+^ in the space close to domains III and IV [15]. When a phenylalkylamine molecule approaches a calcium channel from the cytoplasmic side, its nucleophilic nitrile nitrogen reaches the Ca^2+^, while the other parts of the molecule form affiliated interactions with the amino acids in the P-loop and segments 5 and 6 in domains III and IV of the calcium channel. This causes a physical blockage of ion flow through the channel. In the case of TCCs, there is a lysine (K^3p49^) located adjacent to D^3p50^ in domain III, and the ionized electric potential distribution of the aspartic acid is altered by lysine, which attenuates the electric attraction of aspartic acid (K^3p49^) to Ca^2+^ at the minimum binding distance (4.3 angstroms) and may swing K^3p49^ away from the selectivity filter. Based on this analysis, we suggest that Ca^2+^ will not bind to domain III but to domain I of TCCs. This prediction is consistent with the Ca_v_3.1 structure estimated with cryo-electron microscopy [8], which showed the electron density of the top Ca^2+^ ion is closest to Glu354 of Ca_v_3.1 (E^1p50^, Table 4). Additionally, Rosetta P-loop remodeling shows that the P-loop of domain II is in a more horizontal confirmation than that of other domains, rendering the glutamic acid (E^2p50^) further away from the Ca^2+^ binding site. Since the movements of PAAs will follow the location of Ca^2+^ docking, our models are built for evaluating the affinity of candidate compounds binding to domains I and IV. The compounds that are predicted to have a higher affinity to bind domains I and IV of TCCs but not domains III–IV of LCC (α_1_C) are considered to be ideal selective TCC blocker candidates. This strategy screens out the compounds that are unlikely to bind LCC and TCCs as well as the compounds that are likely to bind both LCC and TCCs. To test whether Ca^2+^ docking is consistent on domains I and II across different TCCs, a molecular dynamics study should be conducted with modified membrane conditions and simulation environments [27].

Previous studies suggest that the nitrile and isopropyl groups in devapamil and some other PAAs serve to guide the drug to the position of Ca^2+^; this function persists if the nitrile is replaced with other elements with high electronegative potentials, such as oxygen or sulfur [15]. In many molecules discussed here, including mibefradil, the nitrile is replaced with a methoxy acetyl side chain or a similar side chain with a high electronegative potential. These molecules behave presumably like those of molecules with nitrile in their alkaline chain. Some of the molecules, such as RO 40-5966 and SKF-96365, do not share the binding mechanism described by our model, and therefore their inhibitory effects on Ca^2+^ channels may not be explained by our new models. For example, by using the input template SKF-96365, we obtained 14 unique structures (as seen in Appendix A), which had negative binding affinities to our TCC models.

Although our models are designed to select compounds that are likely to bind domains I and IV of TCCs, this does not exclude the possibility that PAAs or their derivatives might inhibit TCCs via binding to domains I and II, domains II and III, or even domains III and IV. The goal of our models was to increase the likelihood of success in screening selective TCC blockers based on their chemical structures. 

Our α_1_C model has a similar channel pore size (selectivity filter region) as Ca_v_Ab [9]; however, α_1_G, α_1_H, and α_1_I may have smaller diameters than Ca_v_Ab, since the unitary conductance of TCC currents is smaller than that of LCC. Further statistical analyses of P-loop remodeling data show that a minimal structural difference exists in the P-loop region remodeling data (see Appendix A for details of the statistical analysis).

The Vina screening results of α_1_C identify no binding location for NNC 55-0396 or SKF-96365. The predicted α_1_C binding amino acid for NNC 55-0397, as well as mibefradil, matches the experimental results, which show the inhibitory effect of NNC 55-0397 and mibefradil on LCCs. Although RO 40-5966 has a lower ΔG than mibefradil when binding to α_1_C, the predicted binding location is closer to the center of the channel filter region than mibefradil, which indicates a stronger blocking effect on the rate of Ca^2+^ influx than mibefradil.

Although the K_d_ values predicted by Vina have some gaps compared to the experiment data, they do follow the same order of magnitude (Appendix A). To obtain a more accurate Gibbs free energy for PAAs binding to TCCs, at least two consecutive steps must be conducted: first, the flexible docking process [28]; second, the free energy calculation between ligand and receptor [29]. These two steps require an extensive computational cost and the final K_d_ value is very sensitive to the initial input of the receptor structures. Recently deposited human α_1_G structures offer a good template for developing TCC blockers [8]; however, they have some uncommon regions missing, which could affect PAA binding. Therefore, we argue that it is less likely that the Gibbs free energy of mibefradil/NNC 55-00396 between the prediction and the experiment is matched by choosing different docking programs or conducting a molecular dynamics simulation to find the free energy. 

Our work only focuses on the first step of the drug development process in silico, providing a strategy for predicting the comparative potency of candidate compounds to TCCs versus LCCs. Neither are used for evaluating the pharmacological effects of these compounds on other types of cation channels. Since the strength of the pharmacological effects of PAAs and their derivatives on blocking calcium channels are increased by the appearance of a calcium cation in the channel pore [15], it is unlikely that these PAAs and their derivatives will exhibit a strong inhibitory effect on other cation channels.

## 4. Materials and Methods

### 4.1. Homology Modeling of the α_1_ Subunit

Three classes of calcium channel families have been discovered: Ca_V_1.X, Ca_V_2.X, and Ca_V_3.X. The X represents the subdivisions of the sequence homology of the α_1_ subunit in each class. The models of drug–channel interactions built upon the structural differences in the relevant S5, P-loop, and S6 regions of α_1_G, α_1_H, α_1_I TCCs, and α_1_C LCC, respectively. The protein templates were obtained from BAM [30] using truncated inputs of human α_1_G, α_1_H, α_1_I, and α_1_C (UniProt id: O43497) amino acid sequences (see Table 4 for details). The crystallization structure (PDB id: 5kmh) for the depolarization status of the calcium channel protein, originally extracted from Arcobacter butzleri [9], was employed as the structure template of our model. The multi-domain protein structures of human α_1_G, α_1_H, α_1_I, and α_1_C were built using the ZMM molecular modeling software. The forcefield of specific amino acids was simulated by using the Assisted Model Building with Energy Refinement (AMBER) program. The final structure of the target peptides was optimized by using the Monte Carlo minimization protocol. The maximum iteration time for finding the global minimum was set to 5000. During the energy optimization, structural similarity between target and template was maintained by a flat-bottom parabolic energy penalty function that allows for penalty-free deviations of alpha-carbons up to 1 atom distance from their respective positions in the template, and a penalty was imposed with a force constant of 10 kcal mol-1A-2 for larger deviations [15]. The homology models for human α_1_H (UniProt id: O95180), α_1_I (UniProt id: Q9P0X4), and α_1_C (UniProt id: Q13936) were also built with this method.

### 4.2. Local Electrostatic Potential Calculation

To calculate the electric double layer-related local electrostatic potential while a channel protein interacted with a surrounding water molecule, we generated the corresponding meshes using MSMS (v2.6.1) [31] and set the probe radius to 1.4 and the density to 3.0 for quality control. For truncated amino acid sequences, the local electrostatic potential/binding energy is derived from the summation of the solvation energy and Coulomb energy, i.e., G_complex_ = G_solution_ + G_Coulomb_. The G_Coulomb_ for LCC and TCCs in the P-loop region was calculated by using PyGBe with pre-defined parameters (Appendix A). To compare the influence provided by a single lysine, we used Coulomb’s law to calculate the electric attracting force between Ca^2+^ and aspartic acid:$$F_{\left(\mathrm{Ca},D\right)}={\;k}_{e}\frac{q_{\mathrm{Ca}}q_{D}}{a^{2}},$$
where $k_{e}$ is Coulomb’s constant 8.99 × 109 N·m^2^·C^−2^; a is the distance between Ca^2+^ and aspartic acid; q is the point charge for Ca^2+^, aspartic acid (D), and lysine (K), respectively. The effect of the repellent force on Ca^2+^ by lysine in the direction of the Ca^2+^ and the aspartic acid attracting force is defined by
$$F_{\left(\mathrm{Ca},K\right)}={\;k}_{e}\frac{q_{\mathrm{Ca}}q_{K}}{r^{2}}\cdot \cos\theta ,$$
where θ is the angle between the lines from Ca^2+^ to aspartic acid and from Ca^2+^ to lysine; r is the distance between Ca^2+^ and lysine, which is calculated by
$$r=a\cdot \cos\theta \pm \;\sqrt{b^{2}-\;{\left(a\cdot \sin\theta \right)}^{2}\;}\;,$$
where b is the distance from aspartic acid to lysine. When the angle between Ca^2+^ and aspartic acid, and lysine and aspartic acid (ϕ) is less than 90°, r=a·cosθ +$\;\sqrt{b^{2}-\;{\left(a\cdot \sin\theta \right)}^{2}\;}$; when ϕ is equal to 90°, r = a·cosθ; when ϕ is larger than 90°, r=a·cosθ
$–\;\sqrt{b^{2}-\;{\left(a\cdot \sin\theta \right)}^{2}\;}$. Therefore, the electric force between Ca^2+^ and aspartic acid is
$$F_{\left(\mathrm{Ca},\;D,\;K\right)}\;=\;F_{\left(\mathrm{Ca},D\right)}\;-\;F_{\left(\mathrm{Ca},K\right)}\;=\;{\;k}_{e}\frac{q_{\mathrm{Ca}}q_{D}}{a^{2}}\;-\;{\;k}_{e}\frac{q_{\mathrm{Ca}}q_{K}}{r^{2}}\cdot \cos\theta .$$

### 4.3. Ab Initio Modeling

The ab initio modeling modules from Rosetta were employed to find the three-dimensional structure of target fragmental peptides by sampling and assembling a large candidate pool containing 22,000–27,000 decoy structures for every inputted amino acid sequence [32]. The output results of ab initio modeling were analyzed using the Calibur and energy-based clustering methods.

### 4.4. P-Loop Remodeling

The P-loop region of α_1_G was remodeled using a Rosetta loop modeling module combined with the FastRelax protocol. Twenty-seven amino acids in the P-loop were selected from domains I, II, III, and IV of the TCCs. The modeling used phenylalanine as the starting amino acid and tyrosine, tryptophan, proline, and isoleucine as the ending amino acids. The effective sample size used for subsequent statistical analysis was validated by two groups of data for every remodeled domain. The first group contained 500 output structures, and the second group had 20,000 output structures. The results were analyzed using the Calibur and energy-based clustering methods.

### 4.5. Compound Generation

We used the de novo drug generation package “chemical vae” developed by Gomez-Bombarelli et al. [21] to create a data-driven RNN for new compound production. The dataset we used to train the RNN was prepared by randomly sampling approximately 250,000 compounds from PubChem. The maximum length of encoding for the SMILES-based compounds was set to 120 characters. To analyze the compounds using the RNN, one fully connected layer of width 200 was used. To convert a predicted compound back to the original data type, three layers of gated recurrent units with a hidden dimension of 500 were used. The variational loss of the RNN was annealed according to a sigmoid schedule after 35 epochs, running for 130 epochs while property prediction training the RNN, such that the RNN trained on the PubChem data set with objective properties including: logP, SAS [33], and QED [34]. We kept the other hyperparameters to train the RNN unchanged from the reference [21]. To transfer the predicted compound back to SMILES-based data, we set the Gaussian noise value to 5 and the iteration time to 1000. Once the 2D structure had been obtained, we converted it into a 3D structure via the online program Frog 2.1 [35,36]. The program OpenBabel 2.4.1 [37] was used to add the hydrogen atom and set the pH equal to 7.35 for select compounds.

The 2D structure of mibefradil was employed as a redesigned template for new compounds. Based on its structure, we recreated 129 PAAs and their derivatives. Their corresponding 3D structures (involving up to 800 isomers) were created by Frog 2.1. We used the same program to find the 3D structures for NNC 55-0395, NNC 55-0396, and NNC 55-0397, and combined them with SKF96365 and RO 40-5966, whose 3D structures were downloaded from PubChem, for use as reference compounds for testing and validating the faithfulness of our TCC models.

Some of the candidates of screening compounds may contain oxygen, which replaces the role of nitrile; this structural formula has been reported in certain PAAs such as falipamil, BRL-32872, and tiapamil [15].

### 4.6. Virtual Drug Screening

Virtual drug screening was conducted by using AutoDock Vina [13] with user-defined configuration scripts on the Tulane supercomputer Cypress. The search box was placed in the center of the protein model. The number of mesh elements in the X, Y, and Z directions was set to 60, 124, and 102 for α_1_C and 58, 48, and 50 for α_1_G, α_1_H, and α_1_I, respectively, when simulating the ligand–receptor interaction for existing TCC blockers. The number of mesh elements in the X, Y, and Z directions was increased to 126 when conducting virtual screening for newly designed compounds. To achieve repeatable docking results using Vina, the seed number was fixed at –1460306363. As the grid number for every direction was set to the maximum, Vina had to search a very large three-dimensional space. To find the local minimum, the exhaustiveness was set to 2000 for new compound screening cases and 8 for existing TCC blocker screening cases. The number of predictable binding models expressed as the output was limited to 3 for new compound screening cases and 20 for existing TCC cases.

The Vina output results were checked using PyMOL to ensure the binding locations for existing and newly designed compounds. The predicted binding affinity for the testing compound was calculated as:$$K_{d}=\exp\left(\frac{-\Delta G\;\cdot \mathrm{kcal}\;\cdot {\mathrm{mol}}^{-1}}{0.001986\;\cdot \mathrm{kcal}\;\cdot {\mathrm{mol}}^{-1}\cdot K^{-1}\cdot 310\;K}\right)\;,$$
where ΔG is the Gibbs free energy predicted by Vina.

The 2D ligand–receptor interaction plot was created using LigPlot^+^ [38].

### 4.7. Data Analysis

The Anderson–Darling normality test and the Kruskal–Wallis one-way ANOVA test (Appendix A) were conducted on the generated homology modeling data from Rosetta ab initio modeling and P-loop remodeling in Anaconda Spyder (3.2.8) using a Python 3.6 environment. 

## 5. Patents

All the new identified compounds in this study are patented by the Office of Technology Transfer and Intellectual Property Development at Tulane University (Patent ID: US62/859,519).

## Figures and Tables

**Figure 1 pharmaceuticals-14-00141-f001:**
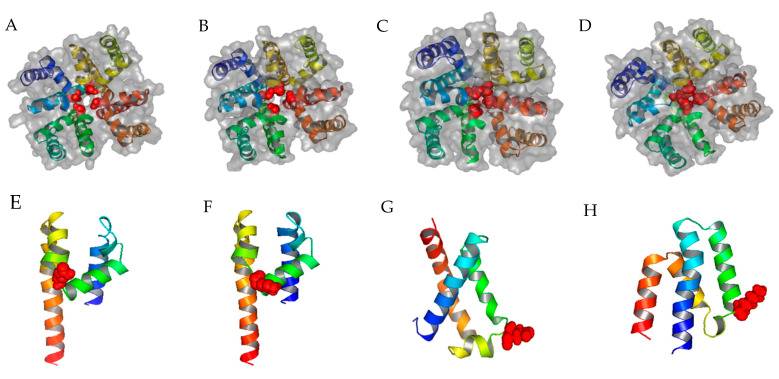
Top views of homology modeling results from ZMM for α_1_C, α_1_G, α_1_H, and α_1_I. (**A**–**D**) The structures of α_1_C, α_1_G, α_1_H, and α_1_I, respectively; blue, green, brown, and yellow represent domains I, II, III, and IV, respectively; the four selectivity-determining amino acids (glutamic acid or aspartic acid) in the P-loop are colored red and displayed as spheres; ZMM generates more consistent 3D structure than the ab initio modeling method for α_1_C and α_1_G; (**E**) the predicted 3D structure of the α_1_C domain III generated by ZMM, the glutamic acid is represented by red spheres; (**F**) the predicted 3D structure of the α_1_G domain III generated by ZMM, the lysine is represented by red spheres; (**G**) the most representative structure selected by Calibur clustering analysis [16] of α_1_C domain III, the glutamic acid is represented by red spheres; (**H**) the most representative structure selected by Calibur clustering analysis of α_1_G domain III, the lysine is represented by red spheres.

**Figure 2 pharmaceuticals-14-00141-f002:**
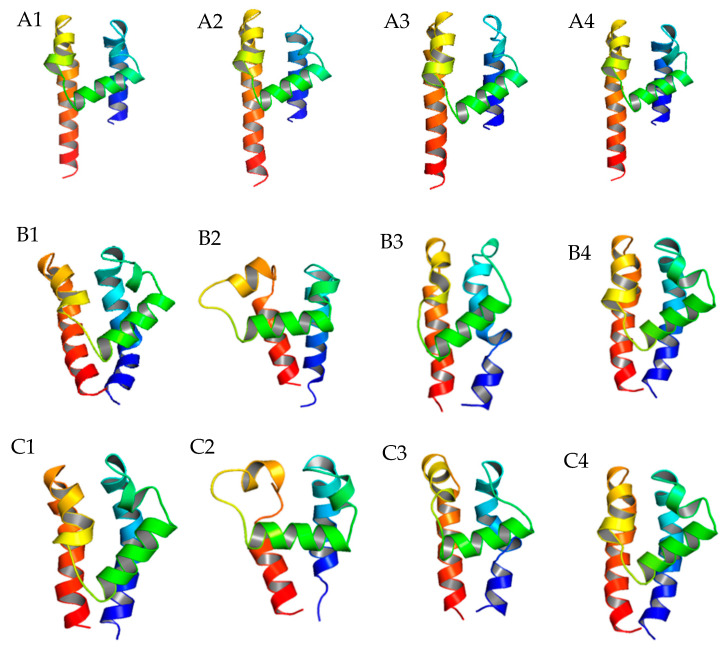
Comparison of the P-loop conformation differences before and after Rosetta P-loop remodeling. (**A**) Homology modeling of P-loop structures of α_1_G domain I (**A1**), domain II (**A2**), domain III (**A3**), and domain IV (**A4**) generated by ZMM; (**B**) Rosetta P-loop remodeling results of α_1_G domain I (**B1**), domain II (**B2**), domain III (**B3**), and domain IV (**B4**) with the sampling size equal to 500; (**C**) Rosetta P-loop remodeling results of α_1_G domain I (**C1**), domain II (**C2**), domain III (**C3**), and domain IV (**C4**) with the sampling size equal to 20,000.

**Figure 3 pharmaceuticals-14-00141-f003:**
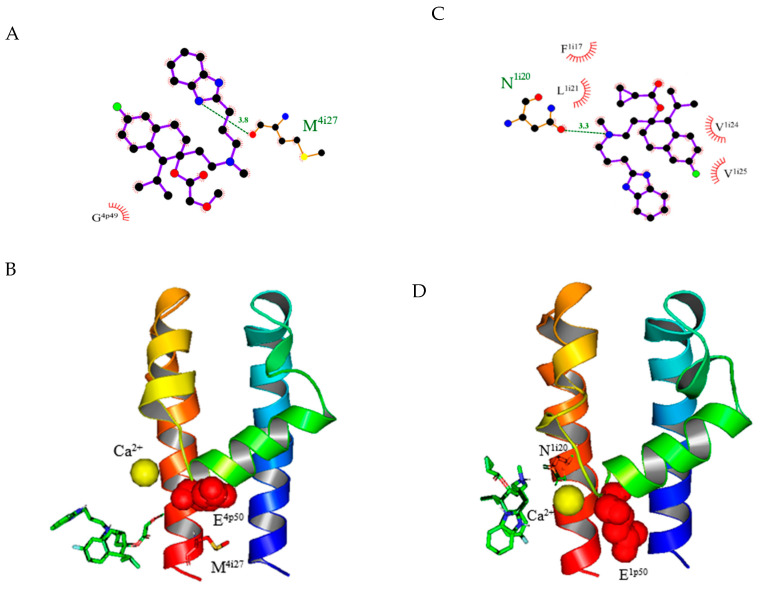
Models of ligand–receptor interactions of mibefradil and NNC 55-0396. (**A**) The predicted binding sites of mibefradil on α_1_C. The H-bond formed between the ammonia (N_3_) on the cyclopentadiene of mibefradil and methionine (M^4i27^) on domain IV. The relative locations of surrounding amino acid residues of the α_1_C L-type calcium channel (LCC) are shown by the arch–dash symbols. (**B**) The predicted 3D binding sites of mibefradil on α_1_C domain IV from a side view. Red spheres represent the position of glutamic acid E^4p50^. Mibefradil is represented by the green ring structure. (**C**) The predicted binding sites of NNC 55-0396 on α_1_G domain IV. The H-bond formed between the central ammonium of NNC 55-0396 and asparagine (N^1i20^) on domain I. (**D**) The predicted 3D binding sites of NNC 55-0396 on α_1_G. NNC 55-0396 is represented by the green ring structure. Red spheres represent the position of glutamic acid E^4p50^. For A and C, carbon, nitrogen, oxygen, and fluorine elements are represented by black, blue, red, and green, respectively; for B and D, the blue and orange ribbon helices represent S5 and S6, respectively. The ribbon helix structures linking S5 and S6 are P-loops. The yellow ball represents the position of the calcium ion.

**Figure 4 pharmaceuticals-14-00141-f004:**
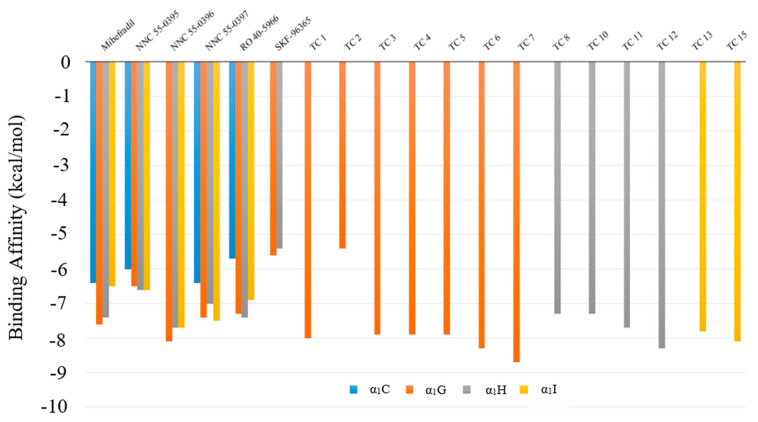
Predicted Gibbs free energy of select T-type Ca^2+^ channel (TCC) blockers and computer-designed compounds of different receptors. Blue—α_1_C, brown—α_1_G, gray—α_1_H, and yellow—α_1_I.

**Table 1 pharmaceuticals-14-00141-t001:** Comparison of numerical results of P-loop electrostatic potential at the four different domains with different lengths of amino acid sequences. TCC: T-type calcium channel.

Channel Domain	Channel Type	AA Sequence Alignment	PyGBe [19] (*E**sol*, *E**co**ul*)
	α_1_G	I T L E G W V D	−11, −407
Domain I	α_1_H	I T L E G W V D	−110, −408
	α_1_I	I T L E G W V E	−116, −409
	Reduced TCC	T L E G W V	−87, −323
	α_1_G	L T Q E D W N K	−217, −631
Domain II	α_1_H	L T Q E D W N V	−262, −631
	α_1_I	L T Q E D W N V	−487, −633
	Reduced TCC	T Q E D W	−164, −425
	α_1_G	A S K D G W V D	−107, −392
Domain III	α_1_H	S S K D G W V N	−113, −425
	α_1_I	A S K D G W V N	−105, −394
	Reduced TCC	S K D G W	−101, −303
	α_1_G	S T G D N W N G	−132, −568
Domain IV	α_1_H	S T G D N W N G	−164, −574
	α_1_I	S T G D N W N G	−177, −574
	Reduced TCC	T G D N W	−86, −393

**Table 2 pharmaceuticals-14-00141-t002:** Predicted Gibbs free energy ΔG of phenylalkylamines (PAAs) on calcium channels (N/A: not available).

Receptor	Drug ID	ΔG (kcal/mol)	Binding Domain
α_1_C	NNC 55-0395	−6.0	IV
	NNC 55-0396	N/A	N/A
	NNC 55-0397	−6.4	IV
	Mibefradil	−6.4	IV
	RO 40-5966	−5.7	IV
	SKF-96365	N/A	IV
α_1_G	NNC 55-0395	−6.5	I
	NNC 55-0396	−8.1	I
	NNC 55-0397	−7.4	I
	Mibefradil	−6.8	I
	RO 40-5966	−7.3	I
	SKF-96365	−5.6	I
α_1_H	NNC 55-0395	−6.6	I
	NNC 55-0396	−7.7	I
	NNC 55-0397	−7.0	IV
	Mibefradil	−7.4	I
	RO 40-5966	−7.4	IV
	SKF-96365	−5.4	I
α_1_I	NNC 55-0395	−6.6	I
	NNC 55-0396	−7.7	I
	NNC 55-0397	−7.5	I
	Mibefradil	−6.5	IV
	RO 40-5966	−6.9	I
	SKF-96365	N/A	N/A

**Table 3 pharmaceuticals-14-00141-t003:** The chemical properties of computer-designed compounds and selected TCC blockers. SAS: Synthetic Accessibility Scores, QED: Quantitative Estimation of Drug-likeness.

Compound Name	logP	SAS	QED
NNC 55-0365	6.8147	3.678636	0.273518
NNC 55-0396	6.0345	3.716436	0.351695
NNC 55-0397	6.2805	3.718535	0.337773
Mibefradil	5.2709	3.71918	0.367183
TC 1	6.1671	4.433975	0.402836
TC 2	5.3351	5.24865	0.408449
TC 3	4.8963	4.777741	0.474028
TC 4	5.0404	4.731275	0.441386
TC 5	5.5879	4.951457	0.415026
TC 6	4.4585	4.79633	0.469604
TC 7	3.6902	4.806084	0.63381
TC 8	6.2697	4.851381	0.242332
TC 10	4.1891	4.542497	0.312353
TC 11	5.372	3.089346	0.276759
TC 12	4.9472	3.971406	0.248549
TC 13	5.7028	4.449589	0.406663
TC 15	4.73	3.921747	0.368816

**Table 4 pharmaceuticals-14-00141-t004:** Amino acid sequences of α_1_G used for searching the homology modeling template.

Channel	Domain/Segment	Residue Label Prefix ^a^	Selected Key Amino Acid Sequence ^b^
			1 11 21
α_1_C	1S5	1o	PLLHIALLVL FVIIIYAIIG LELFMGK
α_1_G	1S5	1o	MLGNVLLLCF FVFFIFGIVG VQLWAGL
α_1_C	2S5	2o	SIASLLLLLF LFIIIFSLLG MQLFGGK
α_1_G	2S5	2o	NVATFCMLLM LFIFIFSILG MHLFGCK
α_1_C	3S5	3o	TIGNIVIVTT LLQFMFACIG VALFKGK
α_1_G	3S5	3o	PIGNIVVICC AFFIIFGILG VQLFKGK
α_1_C	4S5	4o	ALPYVALLIV MLFFIYAVII GMQVFGK
α_1_G	4S5	4o	QVGNLGLLFM LLFFIFAALG VELFGDL
			33 43 53
α_1_C	1p	1p	FDNFAFAMLT VFQCITMEGW TDVLY
α_1_G	1p	1p	FDNIGYAWIA IFQVITLEGW VDIMY
α_1_C	2p	2p	FDNFPQSLLT VFQILTGEDW NSVMY
α_1_G	2p	2p	FDSLLWAIVT VFQILTQEDW NKVLY
α_1_C	3p	3p	FDNVLAAMMA LFTVSTFEGW PELLY
α_1_G	3p	3p	FDNLGQALMS LFVLASKDGW VDIMY
α_1_C	4p	4p	FQTFPQAVLL LFRCATGEAW QDIML
α_1_G	4p	4p	FRNFGMAFLT LFRVSTGDNW NGIMK
			1 11 21
α_1_C	1S6	1i	ELPWVYFVSL VIFGSFFVLN LVLGVLSGEF
α_1_G	1S6	1i	FYNFIYFILL IIVGSFFMIN LCLVVIATQF
α_1_C	2S6	2i	MLVCIYFIIL FICGNYILLN VFLAIAYDNL
α_1_G	2S6	2i	S WAALYFIAL MTFGNYVLFN LLVAILVEGF
α_1_C	3S6	3i	VEISIFFIIY IIIIAFFMMN IFVGFVIVTF
α_1_G	3S6	3i	PWMLLYFISF LLIVAFFVLN MFVGVVVENF
α_1_C	4S6	4i	SFAVFYFISF YMLCAFLIIN LFVAVIMDNF
α_1_G	4S6	4i	VI SPIYFVSF VLTAQFVLVN VVIAVLMKHL

Notes: The difference in amino acid sequences among α_1_G, α_1_H, and α_1_I are underlined. H: L^3o7^, Y4^o16^, R4^o29^, V^2p54^, S^3p47^, N^3p54^, V^2i8^, S^3i15^, A^4i1^, L^4i2^, V^4i5^, T^4i9^, V^4i13^, V^4i23^. I: I^1o19^, V^1p42^, E^1p54^, V^2p54^, P^2i1^, S^2i4^, V^2i8^, L^3o7^, N^3p54^, S^3i15^, Y^4o16^, K^4o29^, Q^4p45^, F^4i1^, V^4i2^, I^4i19^, V^4i23^. ^a,b^ Residue sequences are labeled according to the alignment of the outer helix, the P-loop, and the inner helix of the KcsA structure [15].

## Data Availability

The data presented in this study are available on request from the corresponding author.

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
