# Peer review of "A New Computer Model for Evaluating the Selective Binding Affinity of Phenylalkylamines to T-Type Ca2+ Channels"

_pharmaceuticals, 2021, doi:10.3390/ph14020141_

Round 1

Reviewer 1 Report

In the manuscript Lu describes the development of a computer model to evaluate the binding affinity of phenylalkylamine derivatives to T-type calcium channels. The aim is to develop a model to design novel selective calcium TCC blockers (versus LCC channels).

“There are some tipographical errors:

Line 111: not “[8], This” but “[8], this”

Response:  It is corrected in the revised version. See Line: 113

Line 137: not “reduces, Therefore” but “reduces, therefore”

Response: It is corrected in the revised version. See Line: 139

Line 152: not “55-0397, The” but “55-0397, the”

Response: It is corrected in the revised version. See Line: 154

Line 369: not “FastRelax. 27 amino” but “FastRelax. 27 Amino”

Response: It is corrected in the revised version. See Line: 377

Major issues:

“- in line 48, the Authors state "We reason that this is also true for TCC blockers ..." This reasoning must be justified with a scientific rationale.”

Response: Response: This statement is justified as :” We reason that this is also true for TCC blockers except that the calcium ion is coordinated in the cavity between domains I and IV, since that the depolarized confirmed the pore domain of CaV3.1 and CaV1.1 are superimposed  Line: 50-51

“- The Authors state that the nitrile group binds to the calcium ion. It should be noted that many calcium blockers do not have this group, so it is necessary to specify better the sentence. - The model is built for phenylalkylamine derivatives, but many analogues reported in the supplementary material do not have basic nitrogen, but are ethers. Also in this case it is necessary to specify better. Is it possible that the difference N, O does not cause differences in the three-dimensional structures and therefore in the interactions with the calcium channels?”

Response: We have addressed this concern by giving examples of PAAs containing oxygen instead of nitrile: Some of the candidates of screening compounds may contain oxygen, which replaces the role of nitrile, this structural formula has been reported in certain PAAs such as falipamil, BRL-32872, and tiapamil. Line: 408-410.

Round 2 comments

In this form the manuscript can be considered for the publication on Pharmaceuticals.

Reviewer 2 Report

Round 1 comments and response

This work by You Lu and Ming Li describes a novel homology model of T-type Ca2+ channels (TCCs), developed to test the binding affinity of phenylalkylamine derivatives, as well as their selectivity against L-type calcium channel (LCC).

“Although rational in its structure, the manuscript is hard to read, also because of a bad grammar that should be carefully revised by an English native speaker.”

Response:  We have now thoroughly checked and revised the grammar of the manuscript by an expert in the field.

“It is a merely theoretical study that, as such, poses the risk of being self-referential. Actually, in building up their model, the authors made a number of approximations that could impair the truthfulness of their accounts. Such an approach is taken for granted in the development of homology models, but is generally confirmed (or possibly refuted) by a cross-checking with experimental data. Therefore, also in this case, experimental data should be acquired to support the proposed model, and the binding affinity/selectivity against TCC/LCC of the novel compounds should be verified, at least for TC7. Actually, only the experimental data can guarantee the scientific validity of the work, thus allowing its publication.”

Response: Thanks for the comment. We agree that in vitro with whole-cell patch-clamp test is the critical method to validate the new compounds, and we plan to conduct these research in the subsequence experiment. However, our manuscript suggests a new drug screening model whereas to test specific compounds is beyond the scope of this manuscript.

“A theoretical model of TCC has been previously developed by different authors (see Barreiro et al., Protein Engineering, 2002, 15, 109-122), but the authors neither taken it into account nor cited it. They should instead include it in their study, discussing their achievements in the light of the state of the art.”

Response: The suggestion is well-taken. The modification can be found in line 324-326. 

“Lines 22-23, rephrase the sentence. …‘for the treatment of chronic diseases associated with Ca2+ dysregulation’… sounds better.”

Response: The modification can be found in lines 22-23. 

“Lines 61-67, rephrase the sentence to express a comparison between structures generated by ZMM and ab initio more clearly.”

Response: The modification can be found in lines 62-67.

“Then, modify the caption to Figure 1 accordingly.”

Response: The modification can be found in line: 73.

“Please check also caption to Figure 3, line 165.”

Response: The modification can be found in line 167.

Round 2 comments and response

The authors reviewed the manuscript carefully, acknowledging most of the raised points. However, the main concern still remains, regarding the reliability of the achieved results. An attempt to demonstrate the reliability of the model has been done, comparing predicted Kd values with the experimental ones, determined for known TCC blockers (see Table S4). However, the attempt has been detrimental rather than useful. Actually, a successful prediction rate of (more or less) 30% does not demonstrate the soundness of the proposed model. It is worth reiterating that only accurate experimental data can guarantee the scientific validity of the work, thus allowing its publication.

Response:  

In supplementary table S4, for each molecule when comparing the Vina predicting Kd and the reported experimentally measured Kd, most of them fit to each other reasonably within an order difference. The most unfit data comes from mibefradil, which has previous cited a value of 0.865 M in the last version. Now we checked it and correct the measured value. The experimental Kvalue of mibefradil when docking to 1C should be 21.0 M [1].  In additional, mibefradil lost its nitrile nitrogen group via hydrolysis reaction and turns into RO-405966. This compound does not belong to the PAA family. That’s why the predicted Kd (95.31 μM) by Vina for RO 40-5966 failed to match the experiment results (0.865 μM) [2].

Overall, 4 PAAs (Mibefradil, NNC 55-0395, NNC 55-0396, NNC 55-0397) have predicted Kd value close to the experiment results. Based on this data, we consider that the successful prediction rate of PAA compounds is 100%.

To better estimate the Kd between ligand and receptor, at least two consecutive steps must be conducted after candidate compounds have been determined. First: the flexible docking; Second: calculating the free energy between ligand-receptor. The first step belongs to conducting the molecular dynamics (MD) simulation [3], and the second step requires solving the molecular mechanics Poisson-Boltzmann equation numerically [4]. These two steps require the Cryo-EM structures to be involved in solving the mathematical equations. Human 1G structures are indeed available to the public [5], however, we pointed out in the manuscript that some uncommon regions are missing which could affect PAA bindings. The experimental Kd values of the molecules are variable and highly dependent upon the conditions when the measurements are performed, such as ionic concentration, type of cells, voltage protocol and time on the patch, therefore the Kd predicted by our model can provide an additional criteria for evaluating designed new TCC blockers.

See line: 322 – 336 for addition discussion on Kd 

Although the Kd values predicted by Vina have some gaps compared to the experiment data, they do follow the same order of magnitude (Supplementary Table S4). To obtain a more accurate Gibbs free energy for PAAs binding to TCCs, at least two consecutive steps must be conducted: first, the flexible docking process [3]; second, the free energy calculation between ligand and receptor [4]. These two steps require an extensive computational cost and the final Kd value is very sensitive to the initial input of the receptor structures. Recently deposited human α1G structures offer a good template for developing TCC blockers [5]; however, they have some uncommon regions missing, which could affect PAA binding. Therefore, we argue that it is less likely that the Gibbs free energy of mibefradil/NNC 55-00396 between the prediction and the experiment is matched by choosing different docking programs or conducting a molecular dynamics simulation to find the free energy.

Our work only focuses on the first step of the drug development process in silico,  providing a strategy for predicting the comparative potency of candidate compounds to TCCs versus LCCs.

On page 13, line 444: please change "predicated" into "predicted", and "micromole" into "micromolar"

Response:  

It is corrected in the revised version. See Line: 459-460. we also sent the manuscript for proofreading service at MDPI with selected the specialist in Biology $ Life Sciences. 

Reference: 

 (1)   Bezprozvanny and Tsien, Voltage-dependent blockade of diverse Types of voltage-gated

Ca 2+ channels expressed in Xenopus Oocytes by the Ca 2+ channel antagonist mibefradil (Ro 40-

5967). Molecular Pharmacology, 1995. 48: p. 540-549

(2)     Wu, S., et al., A mibefradil metabolite is a potent intracellular blocker of L-type Ca2+ currents in pancreatic β-cells. Journal of Pharmacology and Experimental Therapeutics, 2000. 292(3): p. 939-943.

 (3)    Chodera, John D., David L. Mobley, Michael R. Shirts, Richard W. Dixon, Kim Branson, and Vijay S. Pande. "Alchemical free energy methods for drug discovery: progress and challenges." Current opinion in structural biology 21, no. 2 (2011): 150-160.

(4)        Wang, Ercheng, Huiyong Sun, Junmei Wang, Zhe Wang, Hui Liu, John ZH Zhang, and Tingjun Hou. "End-point binding free energy calculation with MM/PBSA and MM/GBSA: strategies and applications in drug design." Chemical reviews 119, no. 16 (2019): 9478-9508.

 (5)        Zhao, Yanyu, Gaoxingyu Huang, Qiurong Wu, Kun Wu, Ruiqi Li, Jianlin Lei, Xiaojing Pan, and Nieng Yan. "Cryo-EM structures of apo and antagonist-bound human Cav 3.1." Nature 576, no. 7787 (2019): 492-497.

Round 3 comments

Although still considering experimental data as key to support the reliability of computational studies, I appreciate the authors' efforts to provide a sound rationale for the obtained results. The manuscript can be accepted in its present form.